# Phase transfer catalysts shift the pathway to transmetalation in biphasic Suzuki-Miyaura cross-couplings

Yao Shi [1], Joshua S. Derasp[1] ✉, Tristan Maschmeyer[1] & Jason E. Hein [1,2,3] ✉

The Suzuki-Miyaura coupling is a widely used C-C bond forming reaction. Numerous mechanistic studies have enabled the use of low catalyst loadings and broad functional group tolerance. However, the dominant mode of transmetalation remains controversial and likely depends on the conditions employed. Herein we detail a mechanistic study of the palladium-catalyzed Suzuki-Miyaura coupling under biphasic conditions. The use of phase transfer catalysts results in a remarkable 12-fold rate enhancement in the targeted system. A shift from an oxo-palladium based transmetalation to a boronate-based pathway lies at the root of this activity. Furthermore, a study of the impact of different water loadings reveals reducing the proportion of the aqueous phase increases the reaction rate, contrary to reaction conditions typically employed in the literature. The importance of these findings is highlighted by achieving an exceptionally broad substrate scope with benzylic electrophiles using a 10-fold reduction in catalyst loading relative to literature precedent.

Since its initial reports in 1979[1,2], the Suzuki-Miyaura coupling (SMC) reaction has grown to become an indispensable tool for C-C bond formation, earning a share of the 2010 Nobel Prize in chemistry[3]. Its mild reaction conditions and broad substrate scope have led to applications ranging from large scale pharmaceutical synthesis[4], medicinal chemistry library synthesis[5], natural product synthesis[6], and polymer synthesis[7].

Consequently, the SMC mechanism has attracted substantial attention, particularly regarding the mode of transmetalation for which two major pathways have been proposed[8–29]. The $L_n$Pd(aryl)(X) species could react directly with an 8 electron, 4-coordinate (8-B-4) arylboronate, yielding a Pd-O-B intermediate prior to aryl transfer (Fig. 1, path A). This was an attractive proposal due to the propensity of organoboron species to form 8-B-4 complexes as well as their reactivity in SMC without additional base. However, isolated $L_n$Pd(aryl)(OH) species were reported to rapidly react with 6 electron, 3-coordinate (6-B-3) boronic acids, yielding biaryl products supporting the alternative oxo-palladium pathway (Fig. 1, Path B)[30]. Subsequent studies have aimed to elucidate the favored pathway of SMC transmetalation.

Seminal kinetic studies by Hartwig[13] and Amatore[12] under stoichiometric conditions suggested that $L_n$Pd(aryl)(OH) is the dominant reactive species under their reaction conditions. Stoichiometric studies by Denmark et al. successfully identified and characterized the putative Pd-O-B intermediate using rapid injection nuclear magnetic resonance (NMR) spectroscopy techniques[17,18]. They highlighted that both pathways could lead to the required intermediate; however, path B was more facile under their conditions. In contrast, stoichiometric experimentation reported by Milner and coworkers under biphasic conditions showed similar rates in some cases depending on the nature of the halide used and thus could not conclusively favor either path A or path B under their conditions[14].

Evidence favoring path A had primarily come from a kinetic study by Soderquist, et al.[26] using alkylboron species as well as the computational work conducted by Maseras, Ujaque, et al.[27,28]. In 2014, Santos, Silva, et al. devised a clever study observing product distributions of competing arylboron nucleophiles which supported that path A was active under their conditions[16]. Recent work by Denmark and coworkers provided strong evidence for the direct transmetalation of an 8-

[1]Department of Chemistry, University of British Columbia, Vancouver, BC V6T 1Z1, Canada. [2]Department of Chemistry, University of Bergen, Bergen, Norway. [3]Acceleration Consortium, University of Toronto, Toronto, ON, Canada. ✉e-mail: jderasp@chem.ubc.ca; jhein@chem.ubc.ca

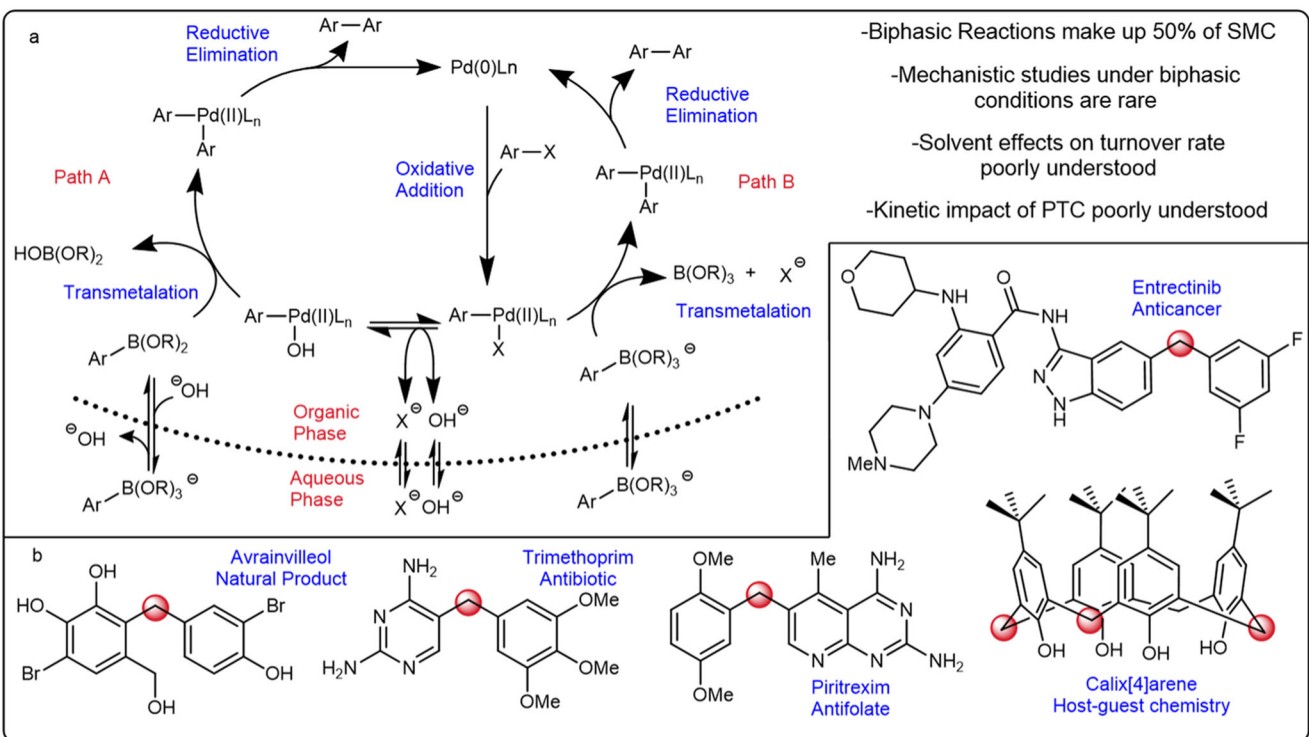

**Fig. 1 | Overview of SMC mechanism and relevant chemical space. a** A simplified catalytic cycle of the SMC under biphasic conditions showing the two commonly accepted paths to transmetalation. **b** Representative examples of the importance of the diarylmethane motif.

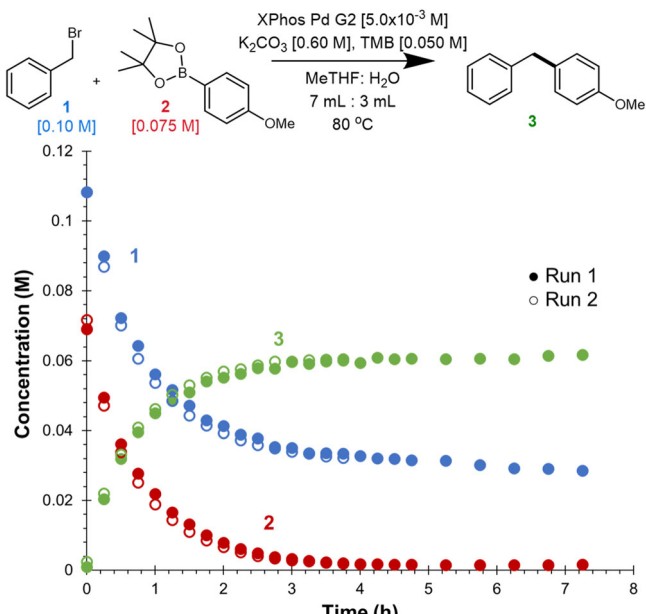

**Fig. 2 | Reproducibility of standard reaction conditions.** Time course overlay of two duplicate runs using the standard reaction conditions.

B-4 complex with $L_nPd(aryl)(X)$ under water-free conditions[21]. The consensus from the literature, as emphasized in recent reviews[8–11], is that both pathways are energetically accessible. The relative dominance of one pathway over the other is expected to depend on the concentrations of each species under the particular conditions employed.

Herein we present a comprehensive mechanistic analysis of the SMC reaction under biphasic conditions. Notably, we observed significant increases in reaction rates when employing phase-transfer

catalysts (PTCs). Investigations of the speciation of both the arylboron nucleophile and the palladium catalyst revealed that the dominant pathway of transmetalation shifts from path B to path A with the incorporation of PTCs. Furthermore, we examined the impact of water content in biphasic systems and found that it has a significantly larger effect on reaction kinetics than does the nature of the organoboron species. This observation challenges the prevailing focus on the organoboron species as a key optimization handle in these systems while water loading is seldom commented upon. These findings were found to be general, enabling an exceptionally broad scope with benzylic electrophiles at low catalyst loadings.

## Results and discussion

We chose to study the SMC under biphasic reaction conditions due to the widespread use of these conditions (~50% of SMCs)[31,32] coupled with the difficulties typically associated with conducting mechanistic studies in such settings. Standard monitoring techniques leveraging infrared (IR) spectroscopy or nuclear magnetic resonance (NMR) spectroscopy have difficulties monitoring biphasic systems. Achieving a homogeneous magnetic field across the sample (i.e. achieving good shimming) is difficult when using NMR for analysis of biphasic systems. Moreover, specialized techniques are required to avoid significant mass transfer effects present in such systems when monitoring static samples. Though IR spectroscopy can be used in systems at high stir rates where mass transfer impacts are alleviated, the changing solvent background provides its own complications. The use of [13]C kinetic isotope effect was used by Joshia et al. and proved a powerful strategy to study the SMC under biphasic reaction conditions[20]. However, this strategy would not lend itself to our ultimate goal of probing the impact of different solvent ratios and additives on the SMC. Manual reaction sampling has also been employed to study the SMC[14,19,22], though reproducibility is typically difficult to achieve in biphasic systems and may explain why these studies tend to limit the proportion of the aqueous phase. Moreover, offline reaction sampling of SMC has been shown to suffer from irreproducibility due to sample aging[29].

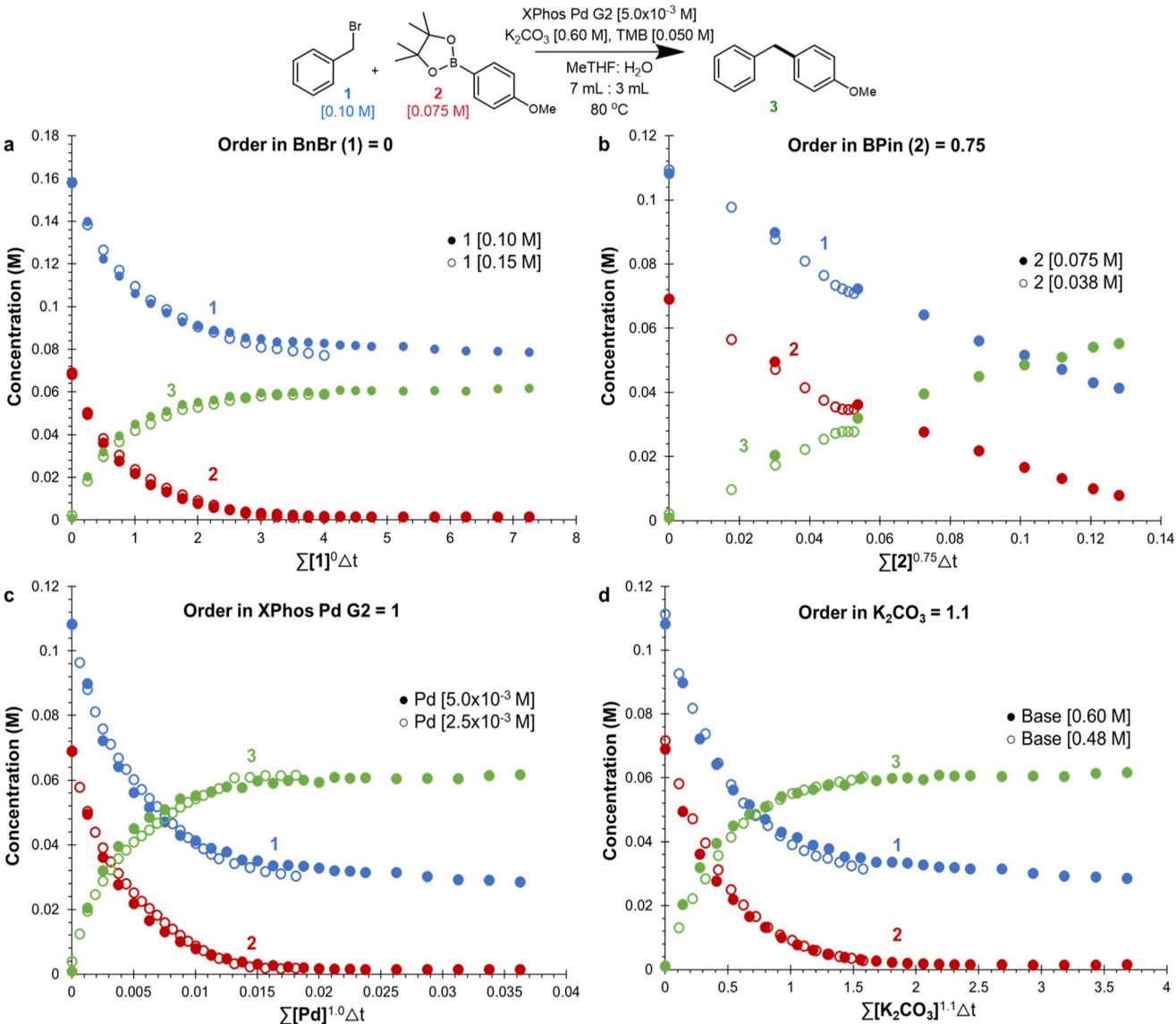

**Fig. 3 | Variable time normalization analysis. a** Different excess of **1**. **b** Different excess of **2**. **c** Different excess of *XPhos Pd G2*. **d** Different excess of $K_2CO_3$.

Recently, we have developed an automated sampling platform which leverages online high-performance liquid chromatography (HPLC) for reaction analysis[33–36]. Notably, this system was shown to provide highly reproducible sampling in heterogeneous settings and thus provides an ideal tool for the current study[33,34].

We chose to target the use of benzylic electrophiles as a model system of study for three reasons; (1) they provide a direct route to the diarylmethane motifs which are broadly present in pharmaceutically relevant compounds[37–50] and supramolecular chemistry[51–58], as well as being important building blocks in their own right (Fig. 1b)[59,60]; (2) mechanistic studies on such systems remains noticeably sparse[61]; (3) the functional group tolerance of these transformations remain noticeably lacking relative to its C($sp^2$)-C($sp^3$) counterpart[62–103]. The prevalence of dialkylbiaryl phosphines in SMCs led us to choose *XPhos Pd G2* as a representative catalyst[104–106]. The use of a weak inorganic base was targeted due to its widespread use in SMC[32]. Thus, the SMC of *benzyl bromide* (**1**) with *4-methoxyphenylboronic acid pinacol ester* (**2**) was conducted in a mixture of 2-methyltetrahydrofuran (MeTHF) and $H_2O$ as the model system.

The reproducibility of the model reaction was probed on our standard automated sampling platform (Fig. 2). Exceptional overlay

between two identical reactions was achieved. Moreover, HPLC allowed us to easily monitor the speciation of **2** to the parent boronic acid under the reaction conditions. Notably, the background rate of hydrolysis of **2** was much slower than the rate of product formation, suggesting transmetalation occurs with **2** directly and not the parent boronic acid (see Supplementary Fig. 4)[25].

With the validity of our reaction monitoring platform confirmed we turned our attention to probing the order of each individual reagent. This will provide an understanding of the turnover limiting step and catalyst resting state, thus providing a foundation upon which to understand the positive or negative impact of solvent changes and additives. We conducted a series of different excess experiments for each reagent and analyzed them using variable time normalization analysis (VTNA) which can be seen in Fig. 3 (see Supplementary Information for complete data set)[107]. The *benzyl bromide* electrophile **1** was not observed to have any impact on the rate of reaction (Fig. 3a).The order in **2** was determined to be 0.75 supporting its involvement in the turnover limiting step (Fig. 3b).The palladium catalyst and the base were observed to have orders of 1.0 and 1.8 respectively (Fig. 3c, d). From these data, a turnover limiting oxidative addition or reductive elimination can be ruled out as the former would

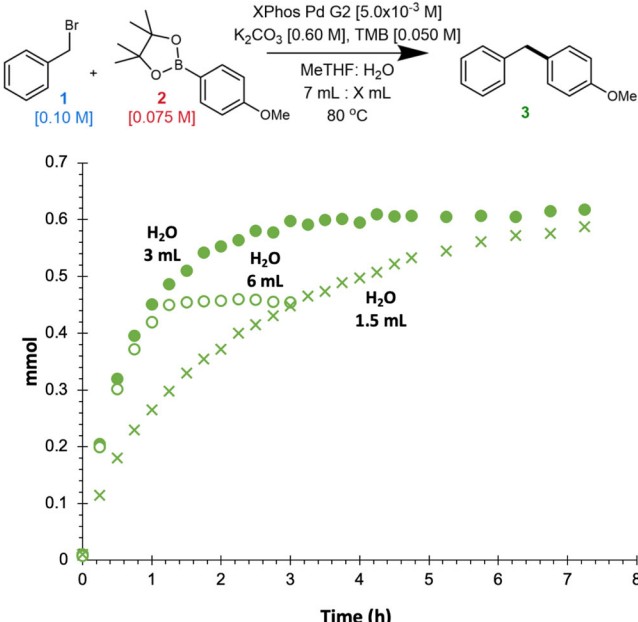

**Fig. 4 | Probing the impact of water.** Time course overlay of **3** probing the effect of different proportions of water while holding the organic phase volume constant. 6 mL ($k_{rel}$ = 1.54), 3 mL ($k_{rel}$ = 1.77), 1.5 mL ($k_{rel}$ = 1). Note: the '6 mL H$_2$O' reaction was conducted on a reduced scale (4.6 mL H$_2$O, 5.4 mL MeTHF) to enable the use of the same reaction flask. This avoids any potential kinetic impacts arising from mixing behavior in different flasks. Consequently, the mmol of starting material and product are lower than the 3 mL and 1.5 mL counterparts.

require a positive order in **1** while the latter would exhibit 0th order behavior in all reagents save for the catalyst. We believe these data are consistent with a rate determining transmetalation coupled with a base mediated pre-equilibrium. A set of same excess experiments were then conducted and confirmed the catalyst suffered insignificant catalyst degradation, though notable halide inhibition was observed (see Supplementary Fig. 13). This halide inhibition was most pronounced with iodide salts, resulting in a dramatic 25-fold reduction in initial rate (see Supplementary Fig. 14). These observations are in line with reports from Milner and coworkers who observed a similar negative impact of added halide salt[14]. These data can be interpreted as pre-equilibrium to transmetalation in which the halide and the base have antagonistic impacts on the former. However, these data do not provide insight on whether the pre-equilibrium is between L$_n$Pd(aryl)(X) and the aryl-boronate (Fig. 1, path A) or L$_n$Pd(aryl)(X) and L$_n$Pd(aryl)(OH) (Fig. 1, Path B).

The significant halide inhibition observed led us to target the development of conditions which would minimize this undesired behavior. Milner and coworkers highlighted that moving from tetrahydrofuran (THF) to toluene in biphasic systems greatly increased reactivity due to the diminished organic phase solubility of the halide salt[14]. However, ethereal solvents are regularly employed in SMC and thus this strategy may not be universal[32]. We hypothesized that increasing the proportion of the aqueous phase in these biphasic systems may decrease the concentration of halide in the organic phase, thus improving reactivity. Despite the widespread use of biphasic reaction conditions, we are unaware of studies probing the effect of this variable on turnover rate. Towards this end, we collected a series of time course reactions in which the volume of the organic phase was held constant, while that of the aqueous phase was altered (Fig. 4). simply increasing the proportion of the aqueous phase resulted in an almost two-fold rate acceleration (see Supplementary Information for discussion of $k_{rel}$ determination). Moreover, if one holds the total volume of the reaction constant while decreasing the proportion

of the organic layer, rate accelerations approaching seven-fold are observed (see Supplementary Fig. 29). This additional benefit is likely attributed to the fact that the reaction primarily occurs in the organic phase of the biphasic system[108]. Thus, decreasing the volume of the organic phase concentrates the reaction despite holding the total volume constant. Overall, these results highlight that simply manipulating the ratio of the aqueous and organic layer can have stunning impacts on reaction rates. To place these results into perspective, the rate increases observed by simply manipulating the ratio of the aqueous and organic phase were larger than the impact of manipulating the identity of the organoboron nucleophile (*vide infra*, Fig. 6) a parameter that has been a central focus in the advancement of the SMC[109–111]. These results stand in stark contrast to the typical biphasic SMC where the amount of water present is typically kept low[31,32].

Although manipulating the solvent composition proved a powerful strategy to improve turnover rates, we wondered if similar improvements could be achieved with the introduction of PTCs. In the case of path A being dominant, a PTC may increase the concentration of the required 8-B-4 species thus pushing the equilibrium towards the Pd-O-B complex. Alternatively, if path B is dominant, a PTC would be expected to push the equilibrium towards the desired L$_n$Pd(aryl)(OH). The use of PTC in SMCs is well documented; however[112], determining its primary role in such systems remains difficult to decipher[108,113]. This arises as a result of the competing effects of PTC in SMCs; (1) the ability to influence the concentration of base/boronate in the organic phase, and (2) the ability of PTCs to stabilize palladium nanoparticles thus minimizing the aggregation, and consequently inactivation, of the palladium catalyst. We were surprised to find that, despite the widespread use of biphasic reaction conditions, the use of PTC as additives remains exceptionally rare outside nanoparticle forming systems.

Our current system of study provides a unique opportunity to probe the impact of PTC in SMCs without the confounding effect of nanoparticle stabilization. We are unaware of any reports in which a Buchwald precatalyst using a dialkylbiarylphosphine was shown to be in a nanoparticle forming regime. Moreover, the catalyst stability observed from the same excess experiment suggests this catalyst system is not prone to deactivation (see Supplementary Fig. 13), thus any impact observed is unlikely due to nanoparticle formation/stabilization.

A series of experiments were conducted using tetrabutylammonium (TBA) salt additives. The addition of 0.10 M *tetrabutylammonium bromide* (TBAB) resulted in complete consumption of the starting material in under 15 min (see Supplementary Fig. 17). Thus, at minimum, the addition of the TBAB salt results in a 12-fold increase in turnover rate. To properly monitor such a rapid reaction, significant modifications to the reaction conditions were made including reducing the catalyst loading, base, and reaction temperature. With these milder conditions, *tetrabutylammonium iodide* (TBAI), TBAB, *tetrabutylammonium chloride* (TBACl), and *tetrabutylammonium hydroxide* (TBAOH) were tested for their ability to promote the reaction (Fig. 5). Notably, speciation of the electrophile (Fig. 5a, inset) and nucleophile (Fig. 5c) of up to 52% and 28% were observed respectively. Given the halide inhibition observed *vide supra* (see Supplementary Fig. 14), this suggests the negative impact of the halide ion must be mitigated by the presence of the TBA counterion. The significant amount of arylboronic acid (**6**) observed in the presence of the TBA salts strongly supports an increased organic phase concentration of hydroxide. Finally, all TBA salts, safe for TBAI, resulted in rate accelerations relative to the standard conditions with no additive. TBAOH provided the highest activity, only slightly outperforming TBACl.

The substantial rate acceleration caused by the addition of TBA salts was believed to arise through increasing the organic phase hydroxide concentration which could significantly impact arylboronate concentration (Fig. 1, path A) and/or catalyst speciation (Fig. 1, path B). Alternatively, one may suspect the hydrolysis of **2** to the more

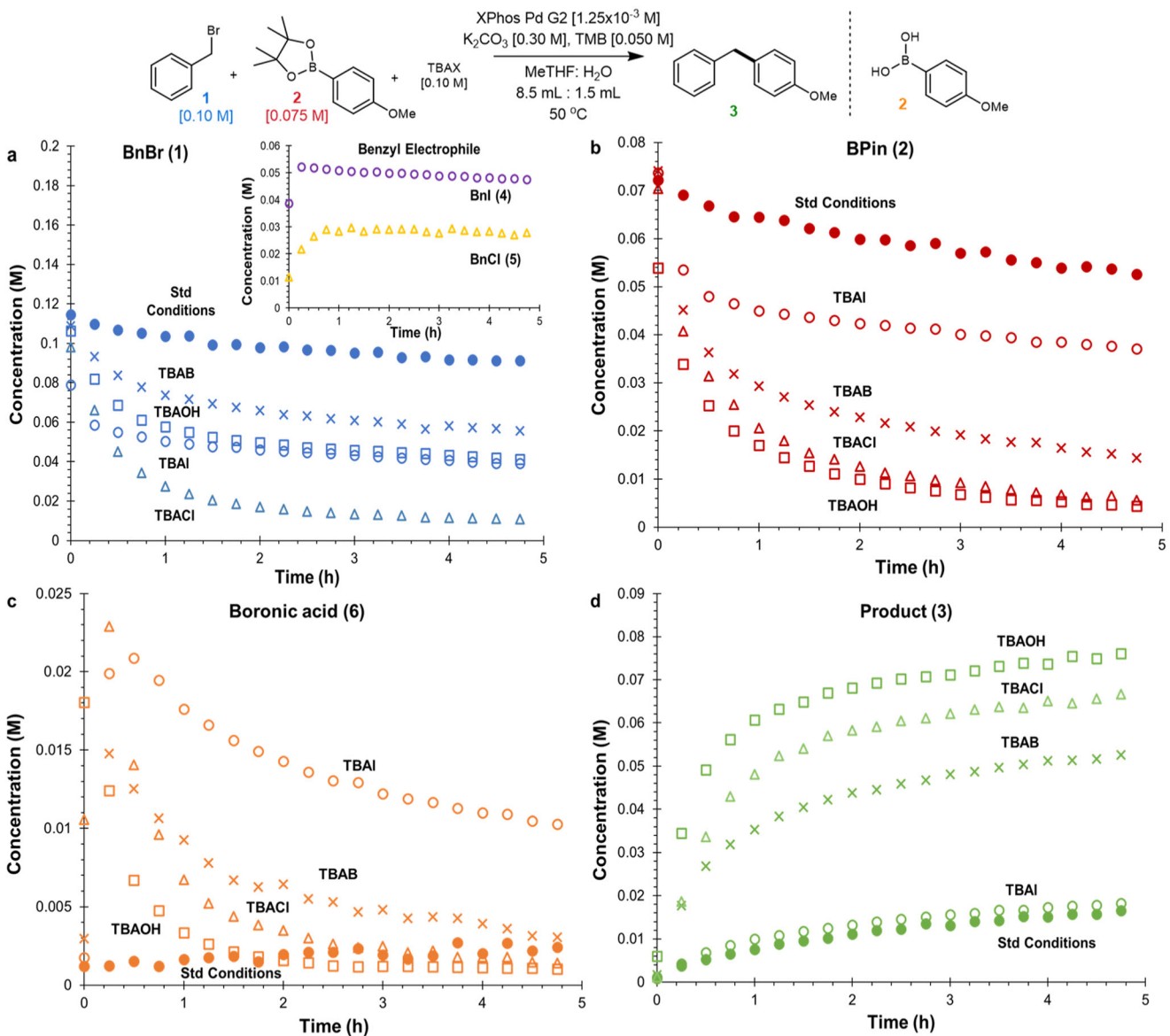

**Fig. 5 | Probing the impact of TBA salts. a** Overlay of **1** with benzyl chloride (**4**) and benzyl iodide (**5**) formation, in the presence of TBACl and TBAI respectively, in the inset. **b** Overlay of **2**. **c** Overlay of boronic acid formation. **d** Overlay of **3**.

reactive **6** could account for such an increase in rate as TBA salts were observed to promote its formation (Fig. 5c). To probe this possibility, we obtained time course data from multiple different organoboron nucleophiles including the parent boronic acid (**6**) (Fig. 6). A neopentyl glycol based boronic acid ester (**7**) had the highest overall reactivity, reaching completion ~2.4 times faster than **2**. Thus, the drastic rate accelerations observed through the introduction of TBA salts are not explained by the differences in reactivity between **2** and **6**. Noteworthy is the comparatively limited impact the nature of the organoboron nucleophile had on turnover rate. While the use of **7** gave just above a 2-fold rate acceleration, the addition of TBAB yielded a 12-fold rate acceleration.

To better understand the mechanism of action of TBA salts, we conducted speciation experiments of **2** and SPhosPd(Ph)(Cl). The choice to use an aryl-derived oxidative addition complex was made due to difficulties in the synthesis of the parent C($sp^3$) oxidative addition complex. Moreover, SPhos was targeted as the ligand as both the hydroxide and chloride adduct have been synthesized, characterized, and employed in such experiments in the past[14,114]. Notably, control experiments revealed the impact of TBA salts is general to the

C($sp^2$)-C($sp^2$) system as well as the alternative ligand (see Supplementary Fig. 31). To probe the palladium speciation, we set up our typical biphasic reaction system with SPhosPd(Ph)(Cl) in the absence of **1** and **2**. The system was equilibrated (see Supplementary Fig. 41) following which a sample of the organic phase was taken and analyzed by $^{31}$P NMR spectroscopy. We focused on the quantities of SPhosPd(Ph)(OH), SPhosPd(Ph)(Cl), and dimeric [SPhosPd(Ph)(Cl)]$_2$ as these will dictate reactivity under the reaction conditions (see Supplementary Information for a detailed discussion). Overall, both monomers were present in similar concentrations with the [SPhosPd(Ph)(Cl)]$_2$ dimer as the dominant species (Fig. 7a). The slight favorability of Pd-Cl complexes is in line with prior reports[13,14]. The same experiment conducted in the presence of TBACl resulted in almost the complete loss of the parent SPhosPd(Ph)(OH) complex with a significant shift towards the formation of Pd-Cl complexes (Fig. 7a). This can be rationalized through the combination of the weak Pd-OH bond strength, coupled with the preference of PTCs to increase the organic phase concentration of the softer chloride anion relative to hydroxide[114].

We then turned our attention to the speciation of **2**. In this case, we set up our standard biphasic reaction system in the absence of **1** and

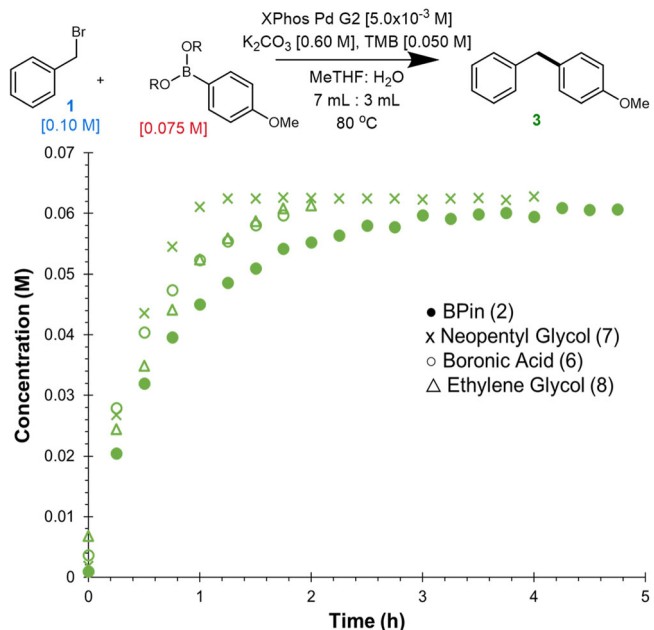

**Fig. 6 | Comparison of different organoboron nucleophiles.** Time course data of **3** comparing the use of **6** ($k_{rel}$ = 1.36), ethylene glycol (**7**, $k_{rel}$ = 1.19), and neopentyl glycol (**8**, $k_{rel}$ = 1.31) to standard conditions ($k_{rel}$ = 1). Note: **7** and **8** hydrolyze to **6** during HPLC analysis (see supplementary information for further discussion).

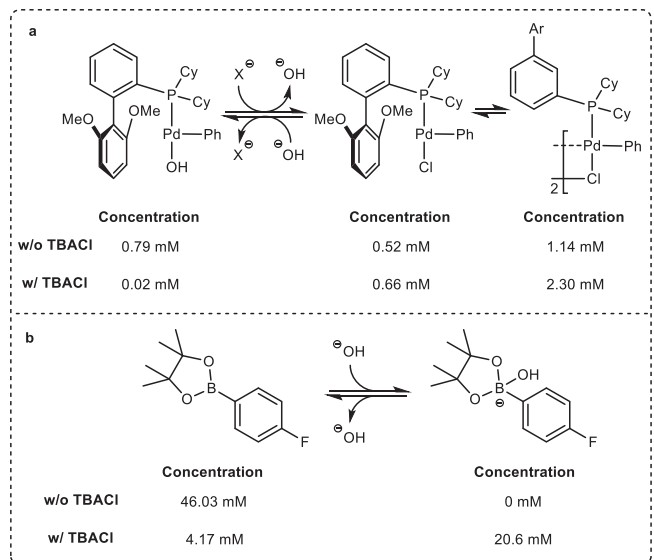

**Fig. 7 | Organic phase Speciation experiments. a** Probing the speciation of the oxidative addition complex with hydroxide and chloride with and without TBACl. **b** Probing the speciation of the nucleophile between the 6-B-3 and 8-B-4 species with and without TBACl.

the palladium catalyst. A sample of the organic layer was taken for analysis by both $^{19}$F and $^{11}$B NMR spectroscopy. Under standard reaction conditions, 8-B-4 species remained below the detection limit in the organic phase of the reaction (Fig. 7b). The parent pinacol ester was the dominant species with small amounts of boronic acid byproduct as expected. In contrast, the presence of TBACl led to the organic phase being dominated by the arylboronate species while some of the parent 6-B-3 species remained (Fig. 7b). This result can be rationalized through the ability of PTCs to increase the organic phase concentration of hydroxide, coupled with the low pKa of arylboron species (see Supplementary Information for a detailed discussion)[115].

Overall, these results suggest that without the TBA additive, the standard conditions are likely dominated by the oxo-palladium pathway (Fig. 1, pathway B). Although both the Pd-OH and Pd-Cl species are present in appreciable amounts, the arylboronate remained below the detection limit in the organic phase. Thus, the positive order in base and negative order in halide salts can be interpreted by manipulating the pre-equilibrium of the catalyst in pathway B while the positive order in palladium and **2** arise from the rate determining transfer of the aryl species onto the catalyst. In contrast, the introduction of TBA salts increases the organic phase concentration of both halides and hydroxides. This results in significant shifts in the speciation of the catalyst towards $L_nPd(aryl)(Cl)$, and **2** towards its 8-B-4 complex. These results suggest that the rate enhancements observed from TBA salts result from shifting the dominant mode of transmetalation from the oxo-palladium pathway (Fig. 1, path B) to the boronate pathway (Fig. 1, path A).

To further validate our conclusion, we carried out a series of different excess experiments while in the presence of TBAB (see Supplementary Figs. 20–27). This revealed the system remained in a turnover limiting transmetalation and thus the rate increases observed are not explained by manipulating the kinetic regime. This was followed by a set of experiments set out to study the impact of halide under these new conditions (Fig. 8). A direct comparison of TBAB versus the use of *tetrabutylammonium triflate* (TBAOTf) reveals a significant positive impact of the bromide under these conditions. This stands in stark contrast to the comparison of KBr versus *potassium triflate* (KOTf). The latter case, as was highlighted earlier, shows a

significant negative impact in the halide ion. This fundamental change in the impact of the halide ion corroborates that a change in the dominant mechanism of transmetalation from pathway B to pathway A occurs in the presence of TBA salts.

Following the completion of our mechanistic study, we wanted to probe the impact of the acquired knowledge on both the catalyst loading as well as the substrate scope of this transformation. The former was probed by testing our model substrate at increasingly diminished catalyst loading (Fig. 9). TBAOH was chosen as the additive as it gave the highest catalyst turnover rate (Fig. 9). Notably, loadings as low as 0.001 mol% were possible while retaining exceptionally high yields. To highlight the scalability of the reaction, as well as its reproducibility at such low catalyst loadings, this reaction was conducted on 45 mmol scale resulting in a 98% yield (Fig. 10, entry **3aa**). A control reaction with no added catalyst confirmed that such reactivity arises from the catalyst and not the presence of trace metals in the starting materials. This result is one of the lowest catalyst loadings reported to date involving benzylic electrophiles[62–103].

We next turned our attention to the substrate scope of this reaction. The use of exceptionally low catalyst loadings (0.1 mol% ≥ ) for the SMCs with benzylic electrophiles has been reported; however, the substrate scope of these reports remains exceptionally limited. To our knowledge, reports where at least one heterocyclic substrate is included range from 1.0 mol% to 10 mol% Pd loading[62–81] with Botella et al.[63] and Chahen et al.[76] having a single example at 0.05 mol% and 0.004 mol% respectively. Thus, we chose to assess the substrate scope using 0.1 mol% catalyst loading. We began by probing the compatibility of different organoboron coupling partners (Fig. 10). Both electron rich (**3ba-da**) and electron poor (**3ea-3fa**) aryl nucleophiles proved competent under these conditions. Heterocyclic pyridyl (**3ga-ha**) nucleophiles produced the product in good yields. The use of thiophene nucleophiles were also well tolerated under the reaction conditions (**3ia-la**) however, in some cases (**3ia-ja**) the competitive formation of *benzyl alcohol* was observed. This issue was largely resolved by replacing TBAOH with TBAB. Pyrrole (**3ma**) and pyrazole (**3na**) derivatives were also well tolerated; however, the latter required the use of slightly higher catalyst loading as well as TBAB to reduce competitive hydrolysis. These conditions are also compatible with the use of a styryl nucleophile with no issues. Finally, we wanted to

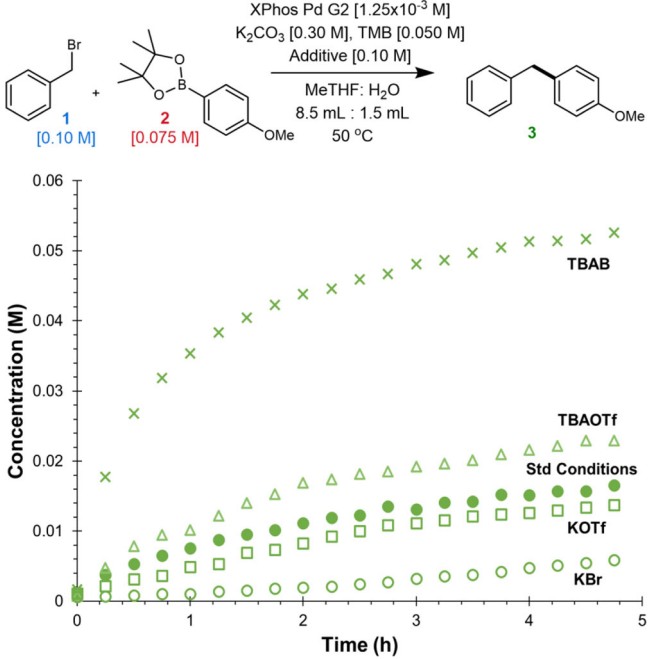

**Fig. 8 | Probing the role of the halide with different salts.** Time course data of **3** probing the effect of TBAB, TBAOTf, KOTf, and KBr compared to standard conditions without these additives.

**Fig. 9 | Probing catalyst loading.** The optimized reaction conditions were repeated with increasingly low catalyst loading.

evaluate the applicability of our conditions on base labile substrates due to the increased organic phase hydroxide concentration in the presence of TBA salts. An organoboron species bearing a methyl ester (**3qa**) or an enolizable ketone (**3ra**) were well tolerated, providing the desired product in an hour with no signs of competitive degradation.

We then turned our attention to the substrate scope of the electrophile (Fig. 10). Both the use of electron poor (**3ab-ad**) and electron rich (**3ae**) coupling partners was well tolerated under the reaction conditions. Both ortho (**3af, 3ai**) and di-ortho (**3ag, 3ah**) substituted benzyl electrophiles were competent coupling partners. Electrophiles bearing Lewis basic pyridyl motifs formed the desired product in good yield (**3aj-3al**). A series of electron rich heterocycles were also well tolerated (**3an-3ao**). Pleasingly, challenging heterocyclic derivatives such as benzothiazoles (**3ap**) and pyrazoles (**3aq**) were compatible without the need to increase catalyst loading. Combining this high reactivity with selectivity in polyhalogenated settings serves as an attractive platform to rapidly generate compound libraries. Towards this end, two electrophiles bearing both $sp^3$ and $sp^2$ halides were targeted. Exceptional selectivity for the $sp^3$ center was achieved in both cases (**3ar, 3at**) in line with prior reports[69]. Furthermore, increasing the loading of the organoboron coupling partner can drive the reaction to the dicoupled product (**3as**). Noteworthy is the compatibility of the phenolic ester of **3at** under the reaction conditions though TBAB was required in place of TBAOH. Finally, we wanted to probe the applicability of further substitutions at the benzylic center. The introduction of a methyl group resulted in modest yield (**3au**). It Is well known that β-hydride elimination can be competitive under such conditions

producing styrene as a dominant side product[67,91]. Our optimized conditions provided the desired product in acceptable yield though styrene was observed. Bromodiphenylmethane derivatives have also been reported to be difficult coupling partners in SMCs[67]. We were pleased to obtain the desired product (**3av**) in modest yield, again highlighting the significant beneficial effect of the incorporation of PTCs.

In conclusion, we have conducted a thorough kinetic study of the SMC with benzylic electrophiles under biphasic reaction conditions. Our automated sampling system provided a robust platform to achieve reproducible reaction sampling despite the biphasic nature of the reaction mixture. A summary of findings in this study is as follows:

1. The addition of PTCs resulted in rate accelerations of up to 12-fold when used as additives. While these additives are commonly used in Pd nanoparticle catalyzed systems for their stabilizing effects, our results indicate that the observed rate enhancements in our system primarily arise from their phase transfer behavior. Thus, these additives should not be overlooked when optimizing biphasic SMC with molecular Pd catalysts which has to date been scarcely employed.
2. The rate accelerations of PTCs were found to be the result of a shift in the dominant mode of transmetalation from the oxo-palladium pathway (without additives), to the boronate based pathway (with additives).
3. Optimizing the solvent ratio when applying biphasic reaction conditions should not be overlooked. Rate accelerations were similar to those observed when changing the nature of the boron species, simply by increasing the volume of the aqueous phase. Further accelerations could be attained by concentrating the organic layer.
4. The significant inhibitory effect of KX salts under standard conditions can be explained by an unfavorable effect on the oxo-palladium pre-equilibrium.

The findings in this study were leveraged to develop a SMC of benzylic electrophiles with an exceptionally broad scope while holding the catalyst loading low. Conceptually, the introduction of TBA salts turns the presence of the halide byproduct from a liability to a beneficial species. We believe these findings will be particularly impactful in the development of telescoped processes or cascades, where significant buildup of halide byproducts is inevitable.

## Methods

### General procedure for reaction time course monitoring

To an oven-dried 15 mL two-neck pear shaped flask with a magnetic stir bar was added 1,3,5-trimethoxybenzene (84 mg, 0.50 mmol), 4-methoxyphenylboronic ester (**2**) (175 mg, 0.75 mmol). If applicable, TBA salts were added at this stage in the reaction flask. The flask was sealed with two rubber septa and put on a Schlenk line. The system was evacuated for 5 min and backfilled with argon three times. Meanwhile, a base solution was made by adding $K_2CO_3$ (1.658 g, 12 mmol) to an oven-dried 9 mL vial. The vial with $K_2CO_3$ was evacuated for 5 min and backfilled with argon three times before 6.0 mL of $H_2O$ was added to the vial. A precatalyst solution was also made by adding XPhos Pd G2 (59 mg, 0.075 mmol) to an oven-dried 3 mL vial. The vial with catalyst was evacuated for 5 min and backfilled with argon three times before 1.5 mL of 2-MeTHF was added to the vial. Then, under a high flow of argon, one septum of the reaction flask was removed and the Easy-Sampler probe was inserted into one neck of the flask. Next, 6 mL of 2-MeTHF and benzyl bromide (**1**) (171 mg, 1.0 mmol) was added to the reaction flask under argon. The reaction was heated to 80 °C and stirred at 1200 rpm. After 15 min at 80 °C, three test samples were taken from the reaction mixture to ensure reproducible sampling. If applicable, potassium salts were added as solids under positive

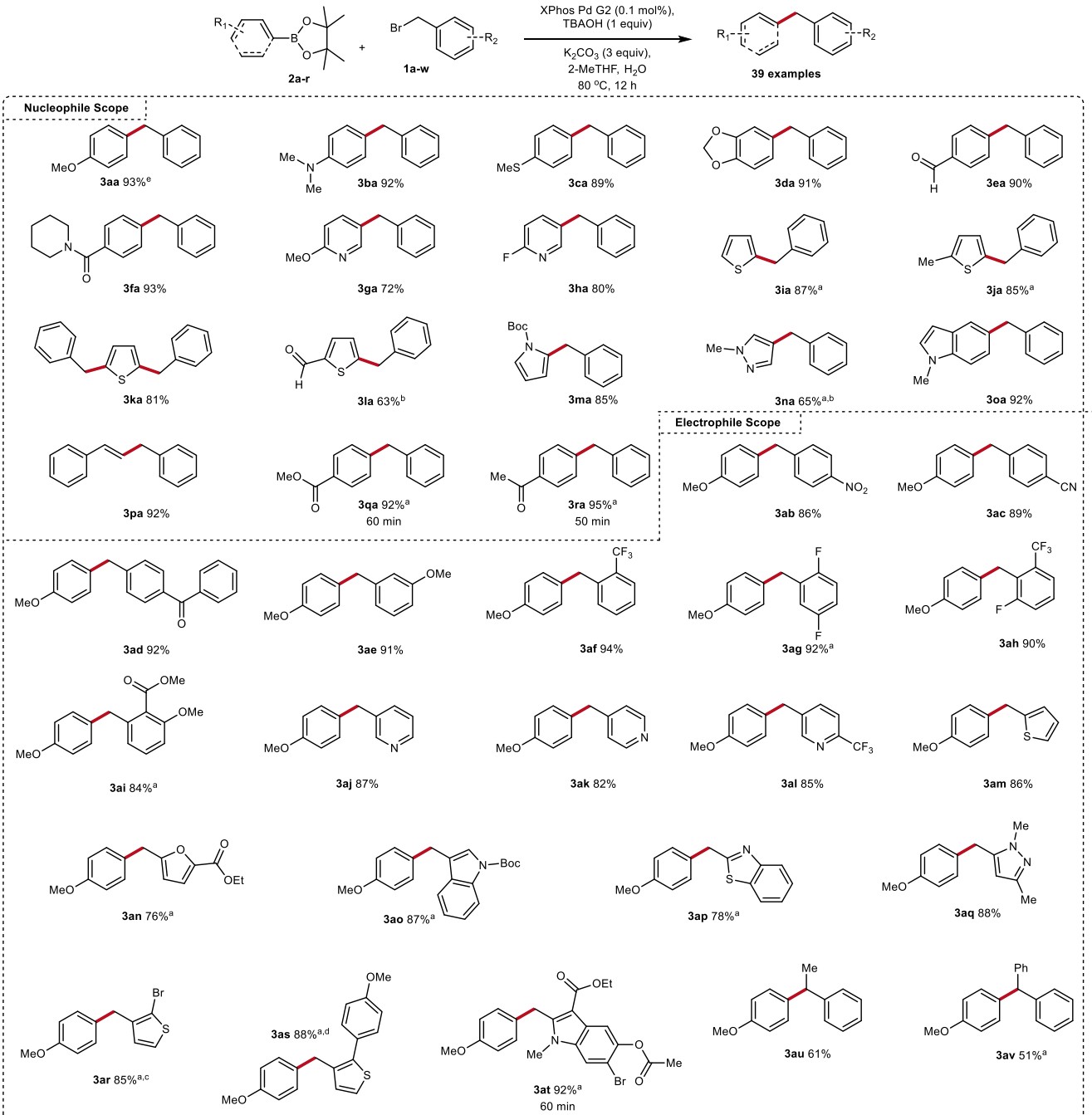

**Fig. 10 | Reaction scope.** Unless otherwise noted, all reactions were carried out on a 0.75 mmol scale with the following conditions: **1a–w** (1.0 equiv), **2a–r** (1.3 equiv), K$_2$CO$_3$ (3.0 equiv), TBAOH (1.0 equiv), 2-MeTHF (1.0 mL), H$_2$O (1.5 mL), 80 °C, 16 h. Isolated yields reported. [a]TBAB instead of TBAOH; [b]1.0 mol% Xphos Pd G2. [c]**1g** (1.3 equiv) and **2a** (1.0 equiv). [d]**1g** (1.0 equiv) and **2a** (2.7 equiv). [e]Large scale reaction: **1a** (1.3 equiv), **2a** (45 mmol), K$_2$CO$_3$ (3.0 equiv), TBACl (1.0 equiv), XPhos Pd G2 (0.001 mol%), 2-MeTHF, H$_2$O, 80 °C, 16 h. Yield determined by internal standard.

pressure of argon at this stage. Finally, 1.0 mL of the catalyst solution, containing XPhos Pd G2 (39.34 mg, 0.05 mmol), 3 mL of K$_2$CO$_3$ base solution were added to the reaction flask and the sampling sequence was immediately started.

**General procedure for a reaction scope entry**
To an oven-dried 3 mL glass vial with a magnetic stir bar, boronic esters (**2a-r**) (1.0 mmol) and TBAOH·3OH$_2$O (800 mg, 1.0 mmol) were added. The reaction vial was sealed with an open top screw cap fitted with a Teflon septum and attached to a Schlenk line via a needle. The system was evacuated for 5 min and backfilled with argon three times.

Meanwhile, a base solution was made by adding K$_2$CO$_3$ (1.658 g, 12 mmol) to an oven-dried 6 mL vial. The vial with K$_2$CO$_3$ was evacuated via a needle for 5 min and backfilled with argon three times before 3.0 mL of water was added to the vial. A precatalyst solution was also made by adding XPhos Pd G2 (7.868 mg, 0.01 mmol) to an oven-dried HPLC vial. The vial with catalyst was evacuated via needle for 5 min and backfilled with argon three times before 1.0 mL of 2-MeTHF was added to the vial. Then, 0.9 ml of 2-MeTHF and the electrophile (**1a-v**) (0.75 mmol) were added through syringe to the reaction flask under argon. Finally, 0.1 mL of the catalyst solution, containing XPhos Pd G2 (0.7868 mg, 0.001 mmol), and 1.5 mL of K$_2$CO$_3$ base were

injected via syringe to initiate the reaction. The reaction was stirred at 1200 rpm in a heat block for 16 h at 80 °C. After cooling, 3 mL of water was added to the mixture and then it was extracted 3 x with 5 mL $CH_2Cl_2$. The organic layer was passed through a $3 \times 1$ cm silica plug, rinsed with 3 mL $CH_2Cl_2$ dried with anhydrous sodium sulfate, and concentrated by rotary evaporation. Reaction products were purified by silica gel column chromatograph or Buchi reverse phase chromatograph system.

## Data availability

The data generated in this study (including characterization of the products, experimental procedures, and detailed kinetic data) are available in the Supplementary Information. All raw and process kinetic data are provided as source data. All other data are available from the authors upon request. Source data are provided with this paper.

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

## Acknowledgements

The authors gratefully acknowledge Mettler-Toledo Autochem for their generous donation of process analytical equipment (EasySampler) to J.E.H. Financial support was provided by Natural Resources Canada (EIP2-MAT-001) to J.E.H. Additional financial support was provided by the University of British Columbia, the Canada Foundation for Innovation (CFI-35883), the Natural Science and Engineering Research Council of Canada (RGPIN-2021-03168) and the Defense Advanced Research Projects Agency (DARPA) for funding this project under the Accelerated Molecular Discovery Program (Cooperative Agreement No. HR00111920027, dated August 1, 2019).

## Author contributions

Y.S. conducted all experimental work. T.M. helped acquire the NMR data for the speciation studies. Y.S., J.S.D. and J.E.H. contributed to devising the project, experimental design and analysis. Y.S., J.S.D. and J.E.H. contributed to the preparation of the manuscript and supplementary information. All authors have given approval to the final version of the manuscript.

## Competing interests

The authors declare no competing interest.
