## [Peer Review File · Nature Communications]

Phase Transfer Catalysts Shift the Pathway to Transmetalation in Biphasic Suzuki-Miyaura Cross-CouplingsReviewer #1 (Remarks to the Author):

Referee report for Hein et al. "Phase Transfer Catalysts Shift the Pathway to Transmetalation in Biphasic Suzuki-Miyaura Cross-Couplings"

In this report, Hein and coworkers conducted a mechanistic study on the Suzuki-Miyaura coupling (SMC) between benzyl halides and arylboronic esters under biphasic, aqueous conditions. Typically, SMCs under such conditions are challenging to study because heterogeneity introduces phenomena related to mass transport; however, the use of online HPLC, a thoughtful analytical approach, enabled reproducible kinetics to be obtained. Using this technique, the authors identified three factors that affect the rate of the reaction: (1) an inhibitory effect by halide salts. (2) a rate enhancement induced by increasing the water content of the reaction (3) a rate enhancement induced by the addition of tetrabutylammonium salts. To explain the significant rate enhancements provided by the tetrabutylammonium salts, the authors examined the speciation of a tangentially related palladium oxidative addition complex and the arylboronic ester by NMR under catalytically relevant conditions, revealing an apparent change in the speciation of both the catalyst as well as the boronic ester. Using these insights, the authors developed a SMC for benzylic bromides and pinacol arylboronic esters with remarkably low catalyst loading.

Throughout the manuscript, good evidence is provided by the authors for the aforementioned effects, often in the form of kinetic studies. Whereas the effect of halide inhibition in SMC reactions has been demonstrated before (see ref 8c), Hein and coworkers clearly differentiate their work by taking a unique approach with regard to their investigation of tetrabutylammonium salts acting as phase-transfer catalysts. Additionally, despite the authors inability to prepare a stable benzylpalladium oxidative addition complex, they provide satisfactory evidence in the Supporting Information to demonstrate the rate enhancements are general with aryl halides and, as such, perform speciation studies with an arylpalladium oxidative addition complex. Interestingly, the results provided by the NMR studies suggest a potential change in mechanism due to a change in catalyst speciation (Path B w/o a tetrabutylammonium salt; path A with a tetrabutylammonium salt), though some of the data presented in the n Supporting Information is lacking in quality, especially some of the ^{31}P and ^{11}B data. Finally, the authors make a claim that the inhibitory effect of KX salts result from a change of active reaction species yet experiments analogous to those performed with tetrabutylammonium salts were not provided.

We chose not to conduct these studies during the course of our experimentation because our conclusions are supported by the work of Milner et al. Both the catalyst system and the biphasic setup targeted along with observed halide inhibition of KX salts from Milner's work shares strong similarities with our work. They went on to demonstrate that, in a THF/water biphasic system, added potassium salt significantly impacts the speciation between SPhosPd(Ph)(Cl) and SPhosPd(Ph)(OH) towards the former (see figure 5 of reference 8c from our manuscript). This finding fits completely the observations delineated within our work where $L_nPd(Ph)(OH)$ is the active species without the addition of TBA salts and thus the negative impact of halide salts can be understood through their influence on palladium speciation towards inactive SPhosPd(Ph)(X).

As for the method that the authors developed, the addition of tetrabutylammonium hydroxide enabled a significantly lower catalyst loading. Whereas the benzylic bromide and pinacol arylboronic ester scope include functionality and substitution patterns that have previously existed in the literature, a few more challenging substrates were demonstrated (e.g. selective coupling with a benzylic bromide over a heteroaryl bromide, two examples of substitution on the benzylic moiety). Ultimately, the preparative advance predominantly enables a very active catalyst with some expansion of the substrate scope.

General Comments

Whereas the inhibitory effect of excess halide salts and the beneficial effect of increased water content is important to document, there is too much detail regarding these studies in the main text, and it ultimately detracts from the focus of the paper: the beneficial effects of tetrabutylammonium salts. Most of these studies could be described in the Supporting Information.

Overall, we agree with the referee that the halide inhibition experiment and part of the experiments probing the impact of water can be moved to the SI. However, we chose to retain the experimental evidence showcasing the impact of increasing the proportion of water while holding the organic phase constant. Given the widespread use of biphasic systems and the pervasive use of limited proportions of water in such settings, we feel it important to highlight this impact in the main text of the manuscript.

Given the amount of kinetic studies shown in the manuscript, when applicable, it would be helpful to have krel shown to help the reader make meaningful comparisons among the datasets.

Additionally, it would help justify some of the hyperbolic language that is pervasive throughout the manuscript (e.g. page 5: "...significant halide inhibition..." when Figure 4 shows that only KI demonstrated significant rate inhibition). This kind of writing detracts from the overall effects that are significant (i.e. the ~12-fold rate enhancements provided by TBAOH).

We agree with the referee that quantifying the impacts on rate using k_{rel} would be helpful to provide comparisons throughout the text and this has been done where appropriate. A lengthy discussion of the validity and limitations of this approach has also been added to the supporting information.

As mentioned previously, conclusions about the nature of the inhibitory effect of KX salts are not explicitly supported, and experiments are needed to demonstrate that it affects the concentration of the oxo-palladium pre-equilibrium.

Please see the above comment

A substantial amount of editing and proofreading are needed before publication, especially within the Supporting Information. Many Supporting Information figures described in the main text are mislabeled in the Supporting Information.

Section Specific Comments:

Page 2:

1. "Milner and coworkers conducted a study under biphasic conditions and obtained kinetic data suggesting both pathways could have similar rates.&c"

a. This statement is a mischaracterization of the data presented in the reference.

Whereas Milner and coworkers do perform a 1:1 stoichiometric experiment with the oxidative addition complex $L(Ar)PdX$ ($X = Cl, OH$) and $PhB(OH)_2/PhB(OH)_3K$, the authors could not deconvolute if path A, path B, or both were occurring.

We agree with the referee and the language with respect to Milner's findings have been modified in the text of the manuscript.

2. "Standard monitoring techniques leveraging infrared (IR) spectroscopy or NMR spectroscopy have difficulties 'locking' onto the changing solvent background."

a. Whereas these difficulties are noted, challenges with mass transport may also be relevant here. This would further strengthen the argument that online HPLC is a unique tool to solve this problem.

We agree with the referee and additional context for mass transfer issues has been provided within the text of the manuscript.

Page 3:

1. The data described by Figures S5-12 are not in agreement with what is written in the text.

According to the Supporting Information, two trials were conducted for VTNA. For example, whereas each trial for the benzyl halide gives an order of 0, the two trials for the

pinacol boronic ester and the base give different numbers (0.60 vs 0.75, 1.3 vs 1.8, respectively). It is unclear why only one replicate is used to represent the order of these two reagents instead of an average. Please provide an explanation.

We had originally tried to obtain orders through overlaying all three different excess experiments on a single plot as the referee suggests however, it became clear that no single order value could achieve acceptable overlay using this approach. Instead, good overlay was achieved using one set of conditions, by sacrificing overlay between the other set. This indicated to us that the order in each reagent at the different concentrations tested are in fact different. It is known that the order of an individual reagent, especially in complex catalytic networks, can change depending on the concentrations targeted. This is described thoroughly in the following reference: *Top. Catal.* 2017, 60, 631. Thus, we decided to determine the order in reagents at each individual concentration. A discussion of this in the text of the manuscript is avoided as the focus of this section is to point towards the turnover limiting step/ catalyst resting state which is not impacted by the specific decisions towards this end.

2. It is quite unusual to have the arylboronic ester as the limiting reagent for an SMC (0.10 M in halide, 0.075 M in boronic ester). Is there a specific reason why the kinetics and downstream experiments were performed this way?

a. Additionally, can the authors justify why the loadings were swapped for general cross-coupling procedure A?

This decision was made at the outset of the project as the stability of the activated electrophiles towards hydrolysis was of concern. However, throughout the course of the project we observed that these species are in fact quite stable towards hydrolysis and instead, homocoupling of the nucleophile is typically formed as a side product. As a result, we chose to conduct the scope of the reaction using the electrophile as the limiting reagent in hopes of providing the highest yields possible.

Page 4:

1. Figure 3:

a. Different excess plots are shown, but these do not indicate clearly to the reader what the order is, especially when quantitative orders are listed in the text. It would be more informative to show the VTNA plots in this figure and move the different excess experiments to the SI.

The manuscript has been updated to provide the VTNA plots in lieu of the time course plots.

2. “2) the low inhibition observed by KCl suggest that chloride-based electrophiles may be ideal when conducting difficult C(sp³)-C(sp²) couplings.”

a. This conclusion is not supported by the subsequent data. Whereas a slight threefold rate enhancement was observed when a benzyl chloride was used instead of a

benzyl bromide, this enhancement does not support the statement.

This part of the text has been removed from the manuscript as a result of other suggestions from this reviewer to condense the discussion around water impacts and halide inhibition.

Page 6:

- a. Figure S27 is incorrectly labeled as "S26" in the text.

The text of the manuscript and SI have been systematically updated to address all inconsistencies with respect to referencing between these documents more broadly.

b. Additionally, Figure S27 mentions that the ethylene glycol boronic ester and the neopentyl glycol boronic ester were hydrolyzed to the boronic acid on HPLC. Was this because of the formic acid additive present in the mobile phase, or do the boronic esters hydrolyze under the reaction conditions? If it is indeed because of formic acid, the data shown in S27 may be different under another analytical technique, as the authors have good data to suggest that the boronic ester, and not the boronic acid, is reacting directly with the oxidative addition complex.

The hydrolysis of these boronic acid esters is not impacted by the presence or absence of formic acid and as such, data on the speciation of these nucleophiles under the reaction condition is not accessible using the current monitoring technique. As the focus of these experiments were to provide evidence that the rate accelerations obtained with TBA salts are not a result of forming high concentrations of the parent boronic acid species in situ, we chose not to develop other monitoring platforms to delineate the speciation of the glycol and neopentyl ester under the reaction conditions as this lies outside the scope of the current manuscript.

c. "Notably, significant amounts of speciation of both the electrophile (Figure 7A, inset) and nucleophile (Figure 7C) were observed."

i. A comparison of how "significant" the boronic acid speciation is would be useful, as the only time it was mentioned prior to this was on page 3.

Percentages with respect to the degree of speciation has been added to the text of the manuscript.

Page 7:

1. Figure 8:

a. See comment regarding Page 6, point b, regarding the deconvolution of the hydrolysis of the neopentyl glycol boronic ester and the ethylene glycol boronic ester. If hydrolysis occurs under the reaction conditions and not because of the HPLC conditions, a statement clearly detailing that the above

boronic esters were hydrolyzed to boronic acids under the reaction conditions should be included.

A statement in the caption of the figure has been added.

Page 8:

1. "To better understand the mechanism of action of TBA salts, we conducted speciation experiments of 2 and SPhosPd(Ph)(Cl) due to difficulties in the synthesis of the parent C(sp³) oxidative addition complex"

a. Whereas it is clear why an arylpalladium species was used instead of a benzylic one, there is no explanation for why the ligand was changed from SPhos to XPhos.

The decision was made use the SPhos derived oxidative addition complexes based on literature precedence of their synthesis, isolation, and use for these studies reported by both Buchwald and Milner while the XPhos oxidative addition complex bound to hydroxide had yet to be reported. Notably, control experiments were conducted to ensure that the TBA additive effects remained relevant in the targeted system. A sentence was added to the text describing our rational for this decision.

2. General comments regarding Figures S31-S38:

a. The organization of the NMR plots is generally confusing. To help deconvolute the information-rich plots for the reader, first, independent standards of ALL relevant species should be prepared and shown and clearly labeled. For example, in S31, the 6-B-3 of the pinacol boronic ester and boronic acid, as well as the 8B4 of the BPin and the 8B4 of the boronic acid should be included. Subsequently, the experimental spectra should be stacked below. This helps the reader clearly deconvolute the effects occurring in each plot.

The NMR data presentation in the supporting information were significantly modified as per the request of the referee to ease their analysis.

b. Secondly, none of the ¹⁹F spectra contain a ¹⁹F reference. Because ¹⁹F chemical shifts are often irreproducible between samples (See Angew. Chem. Int. Ed. 2018, 57, 9528-9533.), it is imperative to have a reference. Thus, the identity of species can be deconvoluted across experiments, ensuring artifacts are not incorrectly identified as important species.

The ^{19}F NMR experiments of the boron speciation experiments were reacquired with the use of an internal standard. No changes from the original data set were recorded. We also used this opportunity to obtain new ^{11}B NMR data using a quartz NMR tube to improve the signal to noise.

c. The NMR experiments claim to be at equilibrium concentrations at 20 mins.

Were further timepoints taken past this point to confirm that the reaction mixture was actually at equilibrium?

Data towards this end was acquired and confirmed that the system had in fact reached equilibrium.

3. Specific comments regarding Figures S31-S38

a. S31:

i. From the text, there is little evidence to suggest that the species that are identified as the 8-B-4 of the pinacolboronic ester (labelled 12) and the 8-B-4 of the boronic acid (labelled 14) are as such.

Independent standards of these species should be prepared to prove their formation in solution.

Obtaining unambiguous characterization of boron speciation and boronate complexes is a highly challenging feat. This is a consequence of the low barrier of oligomerization / ate complex formation which is well known to occur readily under ambient conditions (see: Hall, D. G. Structure, Properties and Preparation of Boronic Acid Derivatives. Overview of Their Reactions and Applications. In Boronic Acids: Preparation and Application in Organic Synthesis, Medicine, and Materials; Hall, D. G., Ed.; Wiley-VCH: Weinheim, Germany, 2011; pp 1–133.). Consequently, if one were to isolate, for example, a boronate complex of an aryl boronic acid, one cannot be sure that upon dissolution that this species does not rapidly speciate and the subsequent NMR used for characterization is in fact misleading. This is to say nothing of the difficulties of isolating boronate complexes of boronic acid esters which are further complicated by competitive hydrolysis. Consequently, we chose to characterize the formation of the two major boronate complexes used in our study using *in situ* characterization by combining analysis via ^{19}F , ^{11}B , and HPLC analysis. A solution of the parent boronic acid **13** was suspended in THF in the presence of 3 equivalent of TBAOH. The resulting ^{11}B NMR spectra revealed a single broad peak at approximately 5 ppm, in line with the formation of an 8-B-4 species. A sample of this solution was injected onto HPLC for analysis which revealed the sole presence of the parent boronic acid. This confirms that the species formed in situ is capable of reverting back to **13** as is expected. Consequently, the resulting ^{19}F NMR spectra, which also reveals the presence of a single peak is assigned as the boronate complex **14**. This exact procedure is repeated with the boronic acid pinacol ester **10** enabling us to characterized its boronate complex. We would like to point out that a similar approach was recently taken by Watson and Lloyd-Jones during their study of boron speciation (Chem. Sci. 2017, 8, 1551) who simply mix the parent boron species in the presence of base and report the new species formed by NMR as the boronate species. Finally, its worth noting that the major conclusion of our work does not hinge on knowing the exact nature of the reactive boron species and instead, relies on the presence/absence of 8-B-4 vs 6-B-3 species which is unambiguously obtained via analysis of ^{11}B NMR.

b. S32:

i. The S/N of the ¹¹B spectrum is extremely poor and attempts to draw conclusions about 8-B-4 formation in this experiment are challenging.

The quality of the ¹¹B NMR were improved as a consequence of reacquiring the NMR data using a quartz tube.

1. It is unclear if the authors ran this experiment in a quartz NMR tube. If not, they should rerun this experiment with one, which will help remove the broad signal observed around ~0 ppm.

See above comment.

c. S35

i. The S/N of the ³¹P spectrum is poor.

To bolster our confidence in the ³¹P NMR spectrum results, the experiments were carried out at an increased concentration to improve the signal to noise and revealed the same trend.

ii. Following the above general comment, independent spectra of 15 and 16 without base would be informative for the reader.

This comment has been addressed.

4. "Under standard reaction conditions, 8-B-4 species remained below the detection limit in the organic phase of the reaction (equation 2). The parent pinacol ester was the dominant species with small amounts of boronic acid byproduct as expected. In contrast, the presence of TBACl led to the organic phase being dominated by the arylboronate species while some of the parent 6-B-3 species remained (equation 2)."

a. When adding a tetrabutylammonium salt, do the following changes in 6-B-3 vs 8-B-4 concentration hold true for the boronic acid and the two other boronic esters examined earlier?

Experimentation with an aryl boronic acid derivatives displayed the same behavior as that observed with the BPin derivative. The fact that these species represent opposite extremes with respect to polarity and steric hindrance (ie: arylboronic acids are highly polar with low steric hindrance at boron, while BPin derivatives are highly non-polar with high steric hindrance at boron) suggest this phenomenon is likely to be broadly applicable no matter the nature of the boron. Furthermore, boronic acid ester derivatives under biphasic conditions hydrolyze to differing degrees to the parent boronic acid and thus the behavior reported in this work is relevant to all biphasic reactions even if certain boronic acid ester derivatives themselves are less so. Consequently, similar experiments towards this end were not carried out on the neopentyl glycol or glycol derivatives.

5. “Thus, the positive order in base and negative order in halide salts can be interpreted by manipulating the pre-equilibrium of the catalyst in pathway B while the positive order in palladium and 2 arise from the rate determining transfer of the aryl species onto the catalyst.”

a. The order in base was nearly two—this statement does not exactly explain this observation.

The order obtained for any individual reagent in a complex catalytic reaction is difficult to interpret without obtaining large quantities of data to obtain an elasticity coefficient (see Bures, *Top. Catal.* 2017, 60, 631.). Consequently, the exact meaning of obtaining an order of 1.8 or 1.1 in base depending on the exact concentrations compared would be speculative. Instead, we focus on the overall impact on reaction rates (ie: base has a positive role, halide has a negative role) and correlate these observations to the measured impact of halide and base on the Pd speciation.

6. Figure 9:

a. “A direct comparison of TBAB versus the use of tetrabutylammonium triflate (TBAOTf) reveals a significant positive impact of the bromide under these conditions. This stands in stark contrast to the comparison of KBr versus potassium triflate (KOTf). The later case, as was highlighted earlier, shows a significant negative impact in the halide ion. This fundamental change in the impact of the halide ion further corroborates a change in the dominant mechanism of transmetalation from pathway B to pathway A occurs in the presence of TBA salts.”

i. The comparison of TBAOTf and TBAB is interesting, but it appears redundant with Figure 8 and Figure 4. If it is not, a more detailed explanation of why this figure is necessary is needed.

We felt this set of data necessary to drive home two points; 1) this allows one to differentiate the impact of the cation from anion highlighting the underlying effect of the halides specifically, and 2) that the impact of the halide in both systems has in fact fundamentally changed under both of these sets of conditions highlighting an underlying mechanistic change.

Page 9:

1. “TBAOH was chosen as the additive as it had resulted in the highest catalyst turnover rate (Figure 7).”

- a. Is base really necessary when using an equivalent of TBAOH?

Yes, we tested the reaction without the addition of the K_2CO_3 and observed diminished yield. This has been added to SI.

2. Table 1:

- a. Table 1 heading is on page 9, but the graphic is on page 10.

This has been addressed in the latest version of the manuscript.

3. “We then proceeded to probe the substrate scope of the organoboron coupling partner.

- a. In the beginning of the paper, comments were made regarding the functional group tolerance of these transformations. Which of these substrates, if any, are specific for this method? Later in the paragraph, two labile examples (methyl ester, enolizable ketone) were presented. Are these the only unique substrates for this method?

We discussed this question thoroughly throughout the course of the project and decided the approach we have taken was the ideal manner in which to highlight the advancements enabled by the findings in our study. The issue with trying to highlight a specific substrate class that was not amenable to the Sp^2 - Sp^3 coupling described is the lack of ‘standardized scopes’ reported in prior works. That is, the literature in this space has largely focused on the application of different and novel catalyst to achieve lower and lower catalyst loadings, with minimal attention paid to the limitations in substrate scope. Thus, it would be a monumental task for us to discover classes of compounds which were simply not amenable to prior art as it would require us to purchase/synthesize the large number of catalyst reported in this space and probe each one with large numbers of substrates to discover which species were in fact not accessible using prior art, as opposed to simply not tested. Instead, we went to great lengths to gather references for all prior work in this space to provide a global comparison of our work vs literature precedence. We believe without question our work provides the broadest substrate scope reported to date in this space with a significant reduction in catalyst loading. Moreover, its worth noting that we focused no effort in trying to identify the most active catalyst system to achieve the remarkable substrate scope reported herein. That is, the choice to us XPhos Pd G2 as the catalyst was simply made based on its widespread use in SMCs and was not chosen as a result of extensive optimization campaign on our end. This means that the results provided in our work are by no means the upper limit of reactivity accessible through the findings reported herein. Indeed, given the large number of designer ligands with exceptional reactivity, significant improvements could be expected when probing ligand effects with the use of TBA additives. Moreover, preliminary data provided in our work suggests the impact of TBA is likely to be broadly applicable to aryl electrophiles as well as other ligand systems.

Page 10:

1. “We then turned our attention to the substrate scope of the electrophile (Scheme 2).
 - a. As mentioned above, are there any challenging coupling partners beyond

the substituted benzyl halides mentioned at the end of the paragraph?

b. In Figure 1, five compounds that contain diarylmethane are shown. Which, if any can be prepared by this method?

Please see the above response.

Final Remarks

Overall, the work is better suited for a more specific audience than Nat. Catal. With the exception of a few confounding experiments, in general, the mechanistic work is strong, and conclusions are well-supported. However, the application of the mechanistic insights for a preparative method is not as convincing. Essentially, the major advance of the method can be distilled down to a reduction in catalyst loading. It is unclear, beyond a few substrates, what advances have been made with coupling partners that were previously inaccessible. If major expansion in the substrate scope were obtained beyond simple benzylic halides, the paper would be very strong.

Reviewer #1 (Remarks to the Author):

The authors performed kinetic experiments using the Direct Inject (DI) system with online HPLC and in situ NMR studies on the Suzuki Miyaura coupling reaction. Direct Inject (DI) system with online HPLC and in situ NMR studies on the Suzuki Miyaura coupling reaction.

In this reaction, the transmetalation step is generally regarded as the rate-limiting step, and the authors investigated whether path A, which proceeds from the LnPd(aryl)(X) species, or path B, which proceeds via the LnPd(aryl)(OH) species, is dominant. In the course of their studies, they also found that quaternary ammonium salts as phase-transfer catalysts have an acceleration effect on the reaction. I think this paper is worthy of publication because it describes a mechanistic study supported by elaborate experiments and the discovery of an acceleration effect based on this study.

Minor

Points

Not all spectra are assigned to compounds, as is the case with species 17 in ^{31}P NMR.

This has been addressed in the supporting information.

It is very impressive that the reaction proceeded at 0.001 mol%, but the reaction scale of the experiment should be described. For a substrate/catalyst ratio of this magnitude, the scale would have to be 10 grams or more to be credible.

To conduct these experiments, we leveraged the use of stock solutions to ensure that the amount of catalyst weighed out for each of these experiments was similar to that used for the time course monitoring. This helps ensure the accuracy and reproducibility of these results despite the exceptionally low loadings of palladium used in the reaction itself. To further support this, a single experiment on 45 mmol scale at 0.001 mol% was conducted and resulted in an even greater yield of 98%.

The position of OH- at the bottom of the arrow in reaction equation (2) is not correct.

We thank the referee for their attention to detail! This has been addressed in the final version of the manuscript.

How likely is it that Pd nanoparticles or Pd black are being produced? Even if Pd nanoparticles are being formed, they may not be affecting the reaction, but I am curious. Dynamic light scattering (DLS) data would be nice, but just a normal photograph would be fine.

We believe the same excess experiments reported herein, coupled with the lack of precedent for nanoparticle formation using our specific catalyst system, strongly support the absence of significant impacts of nanoparticle formation in our system. A picture of our reaction mixture prior to the end of the reaction is provided in the SI to support this assessment as proposed by the referee.

XPhos Pd G2 is not an IUPAC name, so the official name should be written somewhere. Maybe it is written, but it is not immediately obvious. In particular, we need information on whether this compound contains halogens.

This comment has been addressed in the main body of the paper.

Reviewer #3 (Remarks to the Author):

The work is meticulously done and provides detailed mechanistic information of the role of additives to enable SMC in the presence of water- an old process trick-but now mechanistic insights are provided.

This should be well received but where does the Lipshutz work fit in to this story- a comment/footnote might be important here. The Lipshutz surfactants enable fantastical performances. How can this be explained in the context of the present work. Might be worth mentioning somewhere in the paper. We agree with the reviewer that work from the Lipshutz lab has achieved very impressive results using their own additives to achieve reactivity in predominantly aqueous systems. However, it seems that their work has focused largely on 'in water' chemistry and not 'on water' systems which is the focus of our work. Moreover, the nature of the transmetalation pathway operative in these surfactant based systems has not been their focus in contrast to our work where it provides one of the central driving features. Consequently, we have not included a discussion of their work in our manuscript as we felt it lies slightly outside the scope of our work.

Reviewers' Comments:

Reviewer #2:

Remarks to the Author:

Sorry for the delay in peer review.

In response to my comments, I think all points are adequately addressed and the paper is suitable for publication. The presence or absence of Pd nanoparticles cannot be strictly determined from the photo of the reaction solution alone, but it appears to be a clean homogeneous catalyst.

Reviewer #4:

Remarks to the Author:

[Note from the Editor: Reviewer #4 was invited to assess the response given to reviewer #1 who was not able to look over the revision.]

In this contribution the authors use on-line analysis to probe the details of biphasic Pd complex-catalyzed Suzuki-Miyaura coupling, demonstrating the effects of water concentration, halide inhibition and phase transfer catalysis. Moreover, they use these findings to enable lower catalyst concentrations (10 ppm) and broader substrate scope. They have responded in detail to the constructive comments of reviewer 1 especially with regard to their treatment of the VTNA kinetic data and have further clarified their treatment for the reader. In Figure 1, Path A does not account for the fate of X and for Path B the boron product should be B(OH)(OR)₂. Throughout the manuscript they need to change Tf- to OTf- as Tf refers to the -SO₂CF₃ group. The manuscript would also benefit by omission of such platitudes as "to our delight" and "gratifyingly". Finally, they neglected to add Acknowledgements. Several typos/usage suggestions: p. 1, abstract, line 2 - "studies have", "expanding functional", line 4 - "palladium-catalyzed", line 7 - "boronate-based"; text, line 8 - "the SMC mechanism has", line 10 - "for which two"; p. 2, line 8 - "intermediate; however,", line 32 - "employing phase-transfer catalysts (PTCs).", right column, line 2 - "arylboronate, yielding a Pd-O-B intermediate prior to aryl transfer (Figure 1, path A).", line 5 - "than does the", line 19 - "homogeneous"; p. 3, line 11 - "which are broadly", line 12 - "compounds and supramolecular", line 16 - "remains noticeably", right column, line 9 - "see SI for", line 19 - "reagents save for", line 22 - "of 'same excess'", line 25 - "observed (Figure S13).", line 26 - "salts, resulting", line 27 - "rate (Figure S14).", line 31 "transmetalation in which the"; p. 4, line 10 - "phase, thus", line 14 - "reactions in which the", right column, line 2 - "see SI for", line 6 - "observed (Figure S29)."; p. 5, line 14 - "documented; however,", right column, line 12 - "reports in which Buchwald", line 16 - "deactivation (Figure S13); thus", line 19 - "minutes (Figure S17).", line 30 - "supra (Figure S14)."; p. 6, line 2 - "salts is believed", line 18 - "7 gave just:", line 19 - "acceleration, addition of TBAB yielded at"; p. 7, line 1 - "see SI for", line 5 - "aryl-derived", line 12 - "ligand (Figure S31).", line 16 - "equilibrated (Figure S41)", line 19 - "SPhosPd(Ph)(OH) and SPhosPdCl(Ph) and dimeric [SPhosPdCl(Ph)]₂", line 24 - "[SPhosPdCl(Ph)]₂", line 28 - "with a significant", last line (see SI", right column, line 33 - "corroborates that a"; p. 8, line 6 - "it gave the", right column, line 5 - "electrophiles has been", line 7 - "reported; however,", line 11 - "to assess the", line 23 - "tolerated; however", line 27 - "evaluate the", line 31 - "was well"; p. 9, line 2 - "(table 1). Both", line 6 - omit "We then probed the use of heterocyclic benzylic electrophiles", right column, last line - "serves as an"; p. 10, line 15 - omit "as a side product.", line 19 - "yield, again", right column, 6 lines from bottom - omit "TBA, Tetrabutylammonium". In the SI, page S2, lines 8 and 9 they need to add the frequencies for B, F and P NMR, line 20 - "analysis were carried"; p. S6, line 8 - "Figures S6.", line 13 - "methods, suggesting", line 14 - "Excel"; p. S12, line 1 - "study of the competition"; p. S13, line 1 - "check benzyl", p. S14, line 6 - "reactivity of 5'", line 13 - "electrophile, a control"; p. S15, line 2 - "rates; however,", line 10 - "useful; however,", line 15 - "KI; however,", line 19 - "S17 as"; p. S16, line 1 - "Reaction rate comparison" (also on p. S20). On pp. S24, S25, S27-S30, S32, S34, S36, S38 "spectra" should be "spectrum" and the references need to be properly formatted .